# Multimodal Representation Learning Conditioned on Semantic Relations

## Abstract

Multimodal representation learning has advanced rapidly with contrastive models such as CLIP, which align image-text pairs in a shared embedding space. However, these models face limitations: (1) they typically focus on image-text pairs, underutilizing the semantic relations across different pairs. (2) they directly match global embeddings without contextualization, overlooking the need for semantic alignment along specific subspaces or relational dimensions. To address these issues, we propose Relation-Conditioned Multimodal Learning (RCML), a framework that learns multimodal representations under natural-language relation descriptions to guide both feature extraction and alignment. Our approach constructs many-to-many training pairs linked by semantic relations and introduces a relation-guided attention mechanism that modulates multimodal representations under each relation context. The training objective combines inter-modal and intra-modal contrastive losses, encouraging consistency across both modalities and semantically related samples. Experiments on different datasets show that RCML consistently outperforms strong baselines on both retrieval and classification tasks, highlighting the effectiveness of leveraging semantic relations to guide multimodal representation learning.

## 1 Introduction

Multimodal data is increasingly prevalent across domains such as e-commerce, social media, and scientific publishing. Learning unified representations from such data is crucial for enabling understanding, comparison, and generalization across modalities. Due to the lack of large-scale labeled data, contrastive learning became the dominant approach for this goal (Saunshi et al., 2019; Huang et al., 2024), as it learns from weakly paired samples by aligning matched image–text pairs while separating mismatched ones. This paradigm was pioneered by CLIP (Radford et al., 2021), which has since inspired rapid progress along several directions: data-centric scaling, label-supervised extensions, augmentation-enhanced variants, loss reformulations, and modality expansion.

Despite their success, these methods share several key limitations. First, they typically focus on image-text pairs, underutilizing the rich web of semantic relations that naturally exist across samples(e.g., different products, papers, etc.). Second, they match global embeddings directly without contextualization, failing to capture alignment along specific semantic dimensions such as function or style. These challenges are not merely technical, but have tangible impact in real-world use cases. For example, in baby product recommendation, a user focused on infant feeding may consider nursing pillows, milk storage bags, and bottle sterilizers to be closely related. These items span different categories and vary in both appearance and textual description, yet become meaningfully connected under the shared context of infant feeding. Similar issues arise in scientific literature, where content from different papers becomes related under a common methodological theme. On social media, posts about saving money or holiday planning may appear diverse in form but are connected by shared intent. These examples highlight the need to move beyond isolated pairwise contrast, toward modeling sample-level relations within and across modalities to support more contextual and semantically grounded representation learning. While graph-based approaches have also been proposed to capture relations, they largely emphasize graph-level embeddings. In contrast, our work focuses on learning multimodal representations under semantic relations.

To address these limitations, we propose the Relation-Conditioned Multimodal Learning (RCML) framework, which integrates semantic relations between samples into the representation learning

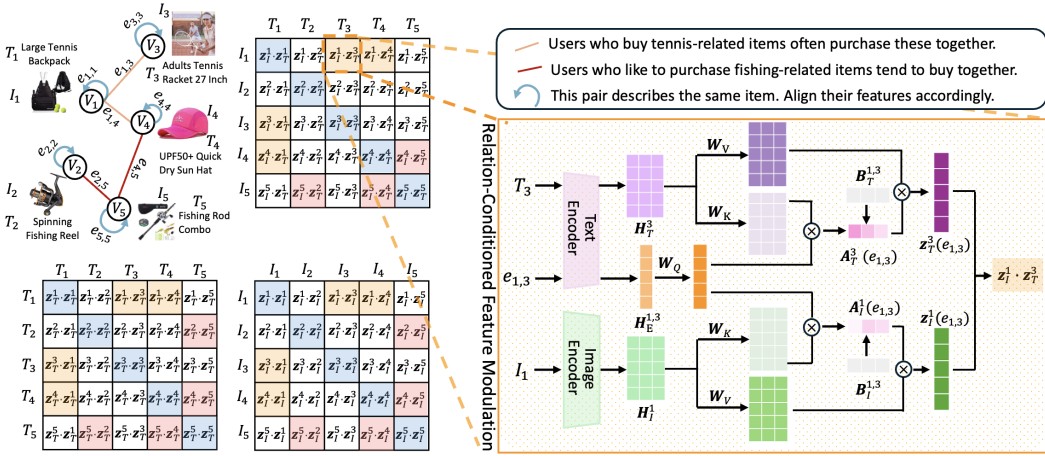

Figure 1: **Overview of the proposed framework.** Each sample consists of a text image pair. Colored elements (red, yellow, blue) represent different semantic relations, which are consistently reflected in sample connections, similarity matrix entries, and feature extraction paths.

process. Instead of relying on isolated image–text pairs, RCML constructs training pairs based on natural-language relations, allowing the model to learn from broader inter-sample structures. It further introduces a relation-guided attention contrastive structure, where relation semantics act as conditioning signals to guide feature interaction and alignment across modalities under specific relational contexts. Finally, RCML includes a contrastive objective that jointly captures cross-modal alignment and intra-modal consistency among related samples. These designs enable RCML to learn contextually grounded and relation-aware representations. Our contributions are summarized as follows: (1) We propose RCML framework that uses semantic relations to guide contextual feature extraction, allowing the model to encode modality-specific information under relational perspectives. (2) We enable contrastive learning across semantically diverse samples by modeling many-to-many inter-sample relations, going beyond traditional pairwise alignment or label-supervised grouping. (3) We conduct comprehensive experiments across two datasets, showing that our method consistently outperforms strong baselines on retrieval and classification tasks.

## 2 RELATED WORK

### 2.1 MULTIMODAL REPRESENTATION LEARNING

Many recent models aim to learn unified representations from multiple modalities such as images and text. Models such as CLIP (Radford et al., 2021) and ALIGN (Jia et al., 2021) train on large-scale web-curated image–text pairs, while later efforts such as LAION (Schuhmann et al., 2022) and DataComp (Gadre et al., 2023) focus on scaling and curating training data at web scale. Subsequent works have explored various ways to improve learning: UniCL and LiT (Yang et al., 2022; **?**) incorporate label supervision to relax strict pairwise alignment, but ultimately still rely on grouping samples under the same class; DeCLIP and SLIP (Li et al., 2021; **?**) use data augmentation and auxiliary objectives to enforce consistency across different views of the same sample, but do not model relations across distinct samples; SigLIP (Zhai et al., 2023) reformulates the contrastive loss using a sigmoid-based objective, and ImageBind (Girdhar et al., 2023) extends joint representation learning to six modalities. Despite their success, these methods generally overlook the semantic relations that exist across different samples. In particular, they lack mechanisms to guide feature extraction under relational context or to align representations across and within modalities based on inter&intra-sample semantics.

### 2.2 GRAPH-BASED MULTIMODAL LEARNING

Graph-based methods have become a prevalent paradigm for modeling structural relationships in multimodal tasks. For example, Qiao et al. (2023)employed GNNs for multimodal sarcasm detection.Liu et al. (2023) leveraged graph structures for multimodal recommendation, and Memon et al. (2025) applied graph-based modeling for information diffusion. Recent works further com-

bine graphs with pre-trained models, such as GraphCLIP (Zhu et al., 2025) for text-attributed graphs and UniGraph2 (He et al., 2025) for unified multimodal graph representation learning. In contrast to these graph-based approaches that rely on GNN-based message passing, our framework is designed as a multimodal feature extraction model, where semantic relations condition the feature extraction process and guide representation learning.

## 2.3 RELATION- AND CONTEXT-CONDITIONED LEARNING

Prompt- and context-based methods provide another way to guide multimodal representation learning. Early works such as CoOp (Zhou et al., 2022) introduced contextual prompts for adapting vision–language models, followed by extensions including, MaPLe (Khattak et al., 2023), PSRC (**?**), and TCP (Yao et al., 2024),MuGCP (Yang et al., 2025) which explore multimodal, semantic-regularized, and task-conditional prompt designs. While effective, these methods mainly perform intra-sample conditioning within single image–text pairs or class contexts. In contrast, our framework conditions feature extraction on inter-sample semantic relations, enabling many-to-many relational modeling beyond task- or class-driven prompting.

## 3 METHODOLOGY

In this section, we first introduce the overall formulation of relation-conditioned contrastive learning, and then describe two key components that support it: relation-guided pair construction and contextual feature modulation.

### 3.1 RELATION-CONDITIONED CONTRASTIVE LEARNING

As shown in Figure 1, each sample is denoted as $V_i = (T_i, I_i)$, where $T_i$ and $I_i$ are the textual and visual descriptions of the item. Pairs of samples $(V_i, V_j)$ are associated with a natural-language semantic relation $e_{ij}$, which serves as contextual guidance for representation learning. When $i \neq j$, $e_{ij}$ encodes inter-sample semantic relation, illustrated by the yellow or red connections in Figure 1. When $i = j$, $e_{ii}$ reflects intra-sample semantic relation, depicted as blue connections in the figure (see Section 3.2 for details). Our goal is to learn an encoder $\mathcal{F}_\theta$ that produces relation-conditioned features $\mathbf{z}_T(e_{ij})$ and $\mathbf{z}_I(e_{ij})$ for each sample under the contextual semantics relation of $e_{ij}$.

Unlike traditional contrastive learning that operates only on matched pairs (i.e., the diagonal of the similarity matrix), our approach supports many-to-many alignment across samples linked by semantic relations. This is visualized in the colored regions of the similarity matrices in Figure 1, where multiple off-diagonal entries are treated as positive pairs, each conditioned on a distinct relation. Different colors correspond to different semantic contexts, indicating that we extract features specifically modulated by the meaning of each $e_{ij}$ rather than relying on a single global embedding.

To optimize such relation-aware alignment, we define a contrastive objective that operates over contextualized features. Beyond cross-modal alignment, we incorporate intra-modal consistency by enforcing that text and image features are coherent under the same semantic relation. Specifically, we design a unified loss with three components: (1) text-to-image and image-to-text contrast for cross-modal consistency, and (2) text-text and (3) image-image contrast for intra-modal coherence. This formulation enables RCML to learn features that are semantically aligned both across and within modalities, all under the conditioning of relation descriptions.The overall loss is given by:

$$\mathcal{L} = \left(\mathcal{L}_{\text{txt-img}} + \mathcal{L}_{\text{img-txt}}\right)/2 + \lambda\left(\mathcal{L}_{\text{txt-txt}} + \mathcal{L}_{\text{img-img}}\right), \qquad (1)$$

where each term follows the same contrastive formulation:

$$\mathcal{L}_{x\text{-}y} = -\sum_{(i,j)\in\mathcal{P}} \log \frac{\exp(\text{sim}(\mathbf{z}_x^i(e_{ij}), \mathbf{z}_y^j(e_{ij}))/\tau)}{\sum\limits_{k\in\mathcal{N}(i)} \exp(\text{sim}(\mathbf{z}_x^i(e_{ij}), \mathbf{z}_y^k(e_{ij}))/\tau)}, \qquad (2)$$

where $x, y \in \{\text{txt}, \text{img}\}$ and $\tau$ is a temperature parameter. Relation-conditioned features $\mathbf{z}_x^i(e_{ij})$ is defined in Section 3.3, and the construction of positive/negative samples is described in Section 3.2.

### 3.2 RELATION-CONDITIONED PAIR CONSTRUCTION

A key component of contrastive learning is the construction of positive and negative sample pairs. In our framework, the positive set $\mathcal{P}$ consists of all sample pairs $(V_i, V_j)$ that are explicitly associated

with a semantic relation $e_{ij}$. These relations are provided as natural-language descriptions and later used to condition feature extraction (Section 3.3). We categorize such positive pairs into two types:

**Intra-sample Relations.** These involve the text and image of the same sample $V_i = (T_i, I_i)$, paired under a generic semantic relation that indicates both views describe the same item. This encourages the model to align textual and visual modalities within a single sample.

**Inter-sample Relations.** These connect different samples $(V_i, V_j)$ through semantic associations such as co-purchase, stylistic similarity, or functional complementarity. Each relation between vertices $v_i$ and $v_j$ comes with textual description $e_{ij}$ which provide semantic meaning of the edge. For example, in recommender system, two items (e.g., product image and product description/review) may be correlated under different contexts, e.g., both bought by "school girls younger than 15", "astronomy enthusiasts", etc. Such context can be expressed as textual descriptions. Similary in book network, textual description can express the context about book relations such as two books under a shared specific tag or user type. Detailed setting of the edge texts for our evaluation data are elaborated in Section 4.1.

The negative set $\mathcal{N}(i)$ for a given anchor sample $V_i$ consists of unrelated samples randomly drawn from the batch that are not paired with $V_i$ under any semantic relation. Each positive pair $(i, j) \in \mathcal{P}$ is contrasted against these negatives using the relation description $e_{ij}$, as defined in equation 2.

## 3.3 RELATION-CONDITIONED FEATURE MODULATION

We utilize CLIP to encode multimodal information for both nodes and edges. Given a sample $V_i \in \mathcal{V}$, its textual tokens $T_i$ and image patches $I_i$ are passed through the CLIP text encoder $f_T$ and image encoder $f_I$, respectively:

$$\mathbf{H}_T^i = f_T(T_i), \quad \mathbf{H}_I^i = \mathrm{MLP}(f_I(I_i)), \tag{3}$$

where $\mathbf{H}_T^i \in \mathbb{R}^{d \times n}$ and $\mathbf{H}_I^i \in \mathbb{R}^{d \times m}$ denote token-level embeddings for the text and image modalities. An MLP projects the image features into the same $d$-dimensional space as the text. For each sample pair with an associated relation description $e_{ij}$, we extract a global semantic embedding $\mathbf{h}_E^{ij} = f_T^{\mathrm{EOT}}(e_{ij}) \in \mathbb{R}^d$ from the EOT token.

While contrastive learning encourages positive pairs to be close in the embedding space, different relations imply different notions of similarity. To capture such contextual variations, we compute relation-conditioned features $\mathbf{z}_x^i(e_{ij})$ for each modality $x \in \{T, I\}$ using an attention-based aggregation mechanism (Vaswani et al., 2017).

$$\mathbf{z}_x^i(e_{ij}) = \mathrm{Norm}(\mathbf{A}_x^i(e_{ij})(\mathbf{W}_V \mathbf{H}_x^i)^\top \mathbf{W}_o), \tag{4}$$

where $\mathbf{H}_x^i$ is the token-level representation, $\mathbf{W}_V, \mathbf{W}_o \in \mathbb{R}^{d \times d}$ are learnable projections, and $\mathrm{Norm}(\cdot)$ denotes L2 normalization. The attention weight $\mathbf{A}_x^i(e_{ij}) \in \mathbb{R}^{1 \times n/m}$ is defined as:

$$\mathbf{A}_x^i(e_{ij}) = \mathrm{softmax}\left((1 - \beta) \cdot \mathbf{q}_E^{ij} + \beta \cdot \mathbf{B}_x^{ij}\right), \tag{5}$$

$$\mathbf{q}_E^{ij} = (\mathbf{W}_Q \mathbf{h}_E^{ij})^\top (\mathbf{W}_K \mathbf{H}_x^i)/\sqrt{d}, \tag{6}$$

where $\mathbf{W}_Q, \mathbf{W}_K \in \mathbb{R}^{d \times d}$ are projection matrices and $\mathbf{h}_E^{ij}$ is the embedding of the relation description $e_{ij}$. The first term provides relation-aware contextual attention by allowing the semantic relation $e_{ij}$ to attend to relevant regions in the modality. The second term $\mathbf{B}_x^{ij}$ is a binary vector that activates only for intra-sample pairs $(i = j)$, highlighting special tokens (e.g., [EOT] or [CLS]) to preserve global consistency. It is designed to capture the inherent alignment between modalities of the same sample, which goes beyond contextual similarity. The coefficient $\beta \in [0, 1]$ balances the influence of contextual relation guidance and undirected alignment.

**Remark.** *CLIP is a special case of our framework with no relation guidance and only cross-modal self-pair training.*

Our framework reduces to the original CLIP formulation under two conditions: (1) The attention reduces to global pooling. When $\beta = 1$ in equation 5, the relation-conditioned attention degenerates to $\mathbf{A}_x^i = \mathrm{softmax}(\mathbf{B}_x^{ij})$, where $\mathbf{B}_x^{ij}$ is a one-hot vector selecting the summary token. This eliminates the influence of $e_{ij}$, and $\mathbf{z}_x^i(e_{ij})$ becomes functionally equivalent to CLIP's global embedding, which pools modality information without contextual semantics.

(2) The objective reduces to pairwise cross-modal contrast. Since $\mathbf{B}_x^{ij}$ is only defined for intra-sample pairs ($i = j$), setting $\beta = 1$ also restricts training to self-pairs. If the contrastive objective is further applied only to cross-modal directions ($x \neq y$), the overall loss reduces to:

$$\mathcal{L} = \left( \mathcal{L}_{\text{txt-img}} + \mathcal{L}_{\text{img-txt}} \right) / 2, \tag{7}$$

where each loss term is computed over same-item pairs:

$$\mathcal{L}_{x-y} = -\sum_{(i,i)\in\mathcal{P}} \log \frac{\exp(\text{sim}(\mathbf{z}_x^i, \mathbf{z}_y^i)/\tau)}{\sum_{k\in\mathcal{N}(i)} \exp(\text{sim}(\mathbf{z}_x^i, \mathbf{z}_y^k)/\tau)}. \tag{8}$$

Together with standard in-batch negative sampling, this configuration recovers the CLIP formulation as a special case of RCML. □

Table 1: Hit@5 (%) for Relation-Guided Retrieval on 8 datasets using five similarity measures. **Bold** numbers indicate the best performance in each dataset.

| Similarity | Elec | Auto | Office | Baby | Pet | Music | Sports | Goodread |
|---|---|---|---|---|---|---|---|---|
| CLIP (TT) | 37.53 | 36.04 | 39.46 | 35.82 | 42.19 | 41.95 | 43.33 | 45.02 |
| CLIP (II) | 31.78 | 27.33 | 34.27 | 29.56 | 34.66 | 34.56 | 35.19 | 32.12 |
| CLIP (TI) | 33.44 | 30.40 | 36.48 | 30.63 | 36.93 | 39.00 | 39.45 | 37.70 |
| CLIP (IT) | 33.81 | 32.14 | 37.92 | 31.62 | 37.95 | 37.51 | 38.82 | 33.84 |
| CLIP (AVG) | 37.08 | 34.01 | 39.14 | 34.66 | 41.34 | 42.59 | 42.87 | 44.24 |
| DeCLIP (TT) | 28.46 | 27.76 | 30.11 | 29.06 | 29.57 | 28.01 | 29.13 | 29.25 |
| DeCLIP (II) | 24.43 | 24.83 | 26.05 | 26.54 | 24.74 | 27.48 | 25.72 | 25.11 |
| DeCLIP (TI) | 23.99 | 22.46 | 23.84 | 23.98 | 22.95 | 24.37 | 24.44 | 27.40 |
| DeCLIP (IT) | 25.00 | 22.38 | 24.02 | 24.03 | 23.55 | 25.22 | 23.83 | 24.05 |
| DeCLIP (AVG) | 27.50 | 27.42 | 29.64 | 30.66 | 28.81 | 29.14 | 29.20 | 28.80 |
| UniCL (TT) | 25.04 | 25.70 | 25.37 | 24.51 | 25.30 | 26.80 | 25.28 | 26.29 |
| UniCL (II) | 31.09 | 29.05 | 35.78 | 30.26 | 35.39 | 34.83 | 34.89 | 33.11 |
| UniCL (TI) | 24.87 | 25.14 | 23.48 | 22.58 | 23.56 | 24.25 | 23.30 | 25.14 |
| UniCL (IT) | 23.74 | 23.96 | 23.02 | 22.47 | 23.56 | 23.24 | 24.37 | 25.35 |
| UniCL (AVG) | 29.92 | 29.18 | 34.03 | 29.04 | 34.58 | 35.59 | 33.02 | 34.41 |
| SigLIP (TT) | 38.91 | 36.63 | 40.79 | 39.91 | 41.19 | 42.41 | 45.78 | 46.42 |
| SigLIP (II) | 36.60 | 30.57 | 37.25 | 33.79 | 37.66 | 38.90 | 41.97 | 35.66 |
| SigLIP (TI) | 36.51 | 33.63 | 38.00 | 34.92 | 40.65 | 42.29 | 46.14 | 37.69 |
| SigLIP (IT) | 36.09 | 33.47 | 38.41 | 34.40 | 38.64 | 41.36 | 45.42 | 34.02 |
| SigLIP (AVG) | 39.69 | 35.15 | 40.71 | 39.17 | 40.76 | 43.14 | 46.62 | 45.22 |
| ImageBind (TT) | 40.25 | 36.73 | 41.96 | 41.53 | 43.73 | 45.94 | 48.09 | 47.20 |
| ImageBind (II) | 38.93 | 31.38 | 39.57 | **36.43** | 39.56 | **41.95** | 42.47 | 38.06 |
| ImageBind (TI) | 38.01 | 35.22 | 39.20 | 35.60 | 41.46 | **44.61** | 46.79 | 39.02 |
| ImageBind (IT) | 38.01 | 34.51 | 38.90 | 35.30 | 40.36 | 41.66 | 45.90 | **34.61** |
| ImageBind (AVG) | 41.63 | 35.68 | 42.94 | 40.13 | 43.77 | 47.07 | 48.48 | 45.25 |
| RCML (TT) | **49.32** | **44.38** | **49.22** | **44.62** | **54.17** | **51.29** | **64.49** | **53.26** |
| RCML (II) | **40.09** | **35.96** | **42.98** | 35.10 | **44.89** | 40.52 | **46.77** | **38.66** |
| RCML (TI) | **42.49** | **37.28** | **43.93** | **38.42** | **48.11** | 41.32 | **52.66** | **39.42** |
| RCML (IT) | **48.00** | **39.77** | **46.33** | **41.20** | **50.00** | **43.50** | **63.36** | 31.43 |
| RCML (AVG) | **49.09** | **44.31** | **49.64** | **44.65** | **55.08** | **51.53** | **64.81** | **49.80** |

## 4 EXPERIMENTS

In this section, we first introduce the experimental setup and baseline models. We then evaluate our framework on three multimodal relation-aware tasks. Finally, we provide detailed analyses of the model's performance and behavior.

### 4.1 EXPERIMENTAL SETUP

We conduct experiments on Amazon Product dataset (Hou et al., 2024) and Goodreads dataset (Wan et al., 2019) (Wan & McAuley, 2018). For Amazon Product dataset, most popular domains such as: Electronics, Automotive, Office Products, Baby, Pet Supplies, Musical Instruments, and Sports are considered. Each product is associated with a title and an image. Pairs of products are considered related if they are co-purchased by users with shared interests, and each relation is annotated with a natural-language description indicating its semantic context. For each domain, we create training and testing sets with no overlap, guaranteeing that both product pairs and individual products are strictly separated. Goodreads dataset shares the same input format as Amazon but defines relations via users' co-reading behaviors. It serves exclusively as an out-of-domain evaluation set, with training restricted to the Amazon domains. Further details of the experimental settings and computing environment are provided in the Appendix A.1 and A.2 .

## 4.2 BASELINE MODELS

To provide a comprehensive evaluation, we compare our method against several strong vision–language baselines. CLIP (Radford et al., 2021) learns aligned image–text embeddings through large-scale contrastive pretraining. DeCLIP (Li et al., 2021) enhances CLIP by introducing data augmentation and auxiliary objectives to improve robustness. UniCL (Yang et al., 2022) incorporates label supervision to unify representations across modalities and domains. SigLIP (Zhai et al., 2023) replaces CLIP's loss with a sigmoid-based formulation to improve data efficiency, and is trained on a larger dataset. ImageBind (Girdhar et al., 2023) extends contrastive pretraining to multiple modalities including audio and depth and we focus on its vision–text component. Notably, it uses a much larger model backbone with significantly more parameters than ours.

## 4.3 DOWNSTREAM TASKS

We evaluate our framework on three tasks designed to test relation-aware multimodal learning under both zero-shot and supervised settings. The first two tasks assess generalization ability without downstream tuning, evaluating whether the model can directly capture semantic alignment guided by relational context. The third introduces a lightweight MLP to examine whether the learned representations are sufficiently discriminative to support supervised relation reasoning.

### 4.3.1 RELATION-GUIDED RETRIEVAL

This task simulates a recommendation-style scenario(Wen et al., 2023; He et al., 2017). Given a source product $A$ and a semantic relation type (e.g., "bought together by people who like fishing"), the goal is to retrieve the most relevant target product $B$ from a candidate set. Each query includes one positive and 20 randomly sampled negatives, forming a 21-way retrieval problem. We report Hit@5 as the primary metric, reflecting realistic recommendation settings where users examine only top-ranked results. For fairness, we concatenate the relation text with the original product text as input to baseline models, ensuring that they have access to the same semantic information.

To compute relevance scores, we extract text and image embeddings using RCML and baselines. We compute five similarities: (1) text-text (TT), cosine similarity between the textual embeddings of $A$ and $B$; (2) image-image (II), cosine similarity between their image embeddings; (3) text-image (TI), from $A$'s text to $B$'s image; (4) image-text (IT), from $A$'s image to $B$'s text; and (5) average (AVG), cosine similarity between averaged text and image embeddings of each product. These scores rank candidates and assess how well each model aligns multimodal features under relational context.

As shown in Table 1, our proposed RCML consistently outperforms all baselines across most settings, achieving the best results on 36 out of 40 metrics. This highlights its strong ability to leverage both multimodal content and relational semantics for context-aware feature extraction, leading to superior retrieval performance. Compared to the standard CLIP backbone, RCML improves overall Hit@5 by approximately 30.79%. While DeCLIP and UniCL are also trained with contrastive objectives, they perform poorly on our relation-targeted retrieval task. DeCLIP emphasizes local consistency, and UniCL relies on label-level supervision—neither captures contextual alignment across semantically related samples. SigLIP performs relatively well, as its sigmoid-based objective enables more flexible pairwise alignment, while ImageBind benefits from a powerful image encoder and large model capacity, which explains its occasional advantage on image-dominant metrics. However, without explicit relation conditioning, both still underperform RCML overall. Furthermore, RCML generalizes well to the out-of-domain Goodreads dataset, highlighting its robustness across different domains of relational data.

### 4.3.2 RELATION TYPE PREDICTION

In this task, the model is given a product pair $(A, B)$ and must identify the most likely semantic relation connecting them from a predefined set of relation types (10 for Amazon domains, 8 for Goodreads). For each candidate relation, we compute the similarity between $A$ and $B$ under the corresponding relation-conditioned embedding, and select the one with the highest score. Since baseline models do not support relation-specific encoding, we report results only for RCML, using five similarity variants across unimodal and cross-modal configurations. We report Top-3 Accuracy on all datasets where relation types are sufficiently meaningful to support evaluation.

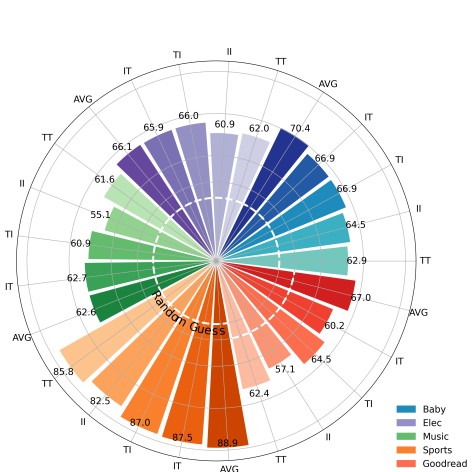

Figure 2: Top-3 accuracy for Relation Type Prediction across five similarity types.

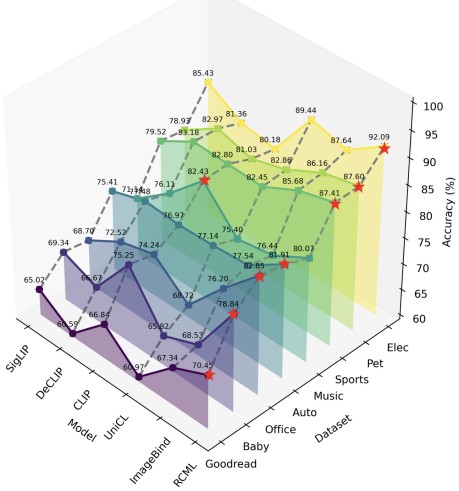

Figure 3: Accuracy on Relation Validity Prediction. ⋆ denotes the best performance.

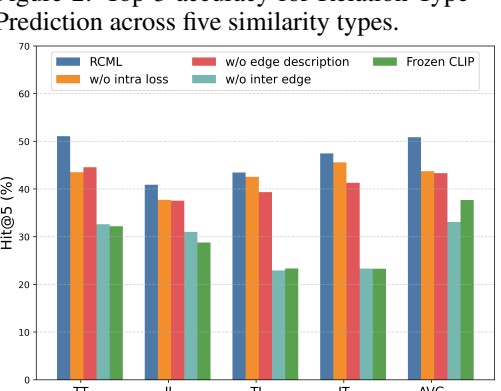

Figure 4: Ablation Results Across Similarity Types on Relation-Guided Retrieval.

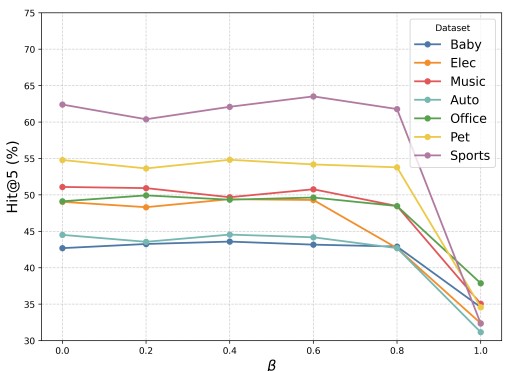

Figure 5: Sensitivity analysis of the attention coefficient $\beta$ on Relation-Guided Retrieval.

As shown in Figure 2, RCML demonstrates strong performance on both Amazon and Goodreads domains, consistently exceeding the random baseline by a substantial margin (30% for Amazon datasets and 37.5% for Goodreads). Among five similarity variants, the averaged configuration performs best overall, suggesting that combining textual and visual cues benefits relation inference.

### 4.3.3 RELATION VALIDITY PREDICTION

Unlike the previous task, which evaluates zero-shot selection among candidate relation types, this task focuses on supervised validation of specific relation instances. Given a product pair $(A, B)$ and a candidate relation type, the model must predict whether a relation of that type exists between them. This is formulated as a binary classification problem, and we report classification accuracy as the evaluation metric. For fair comparison, all baseline models are kept frozen, and a lightweight linear classifier is trained on top of their extracted features. The input to the classifier is the concatenation of text and image embeddings from both products, along with the embedding of the relation label. This setup evaluates whether the learned multimodal representations can support relation-aware prediction when supervision is available.

As shown in Figure 3, RCML achieves strong performance across the evaluated domains, including the out-of-domain Goodreads dataset, demonstrating its robustness when relation-aware supervision is available. While models like CLIP show large domain variance and SigLIP or ImageBind benefit from scale and soft supervision, none incorporate explicit relation-aware mechanisms, leading to weaker performance on tasks requiring fine-grained relational reasoning.

## 4.4 FURTHER EVALUATION AND ANALYSIS

**Ablation Studies.** To evaluate the contributions of different components in our framework, we conduct ablation experiments on the Relation-Guided Retrieval task. Results are averaged over the seven datasets and reported across five similarity types, as shown in Figure 4. We compare five settings: (1) **w/o inter edge**: removes all inter-sample relations, resulting in the largest performance drop (38.68%), which confirms the importance of many-to-many learning across samples. (2) **w/o intra loss**: disables the intra-modal contrastive objective while retaining cross-modal training. The performance drop (8.50%) suggests that intra-modal alignment contributes meaningfully, though it is not the dominant factor. (3) **w/o edge description**: removes the semantic content of relations but retains relation connectivity. The decline (11.60%) highlights the value of contextual guidance in learning relation-aware representations. (4) **Frozen CLIP**: freezes the CLIP encoders and trains only the attention module. The significant drop (37.93%) shows the necessity of end-to-end adaptation for relation-aware learning. (5) **RCML**: our full model achieves the best performance, demonstrating the effectiveness of leveraging semantic relations through many-to-many contrastive supervision. These trends hold consistently across all imilarity types, indicating that each component of our framework contributes positively to both unimodal and cross-modal alignment under relational context.

**Sensitivity Analysis.** As defined in equation 5, the coefficient $\beta \in [0, 1]$ controls the trade-off between relation-specific contextual attention and undirected alignment via global summary tokens. We evaluate the impact of this balancing coefficient in Figure 5. Model performance remains relatively stable across a broad range of $\beta$ values (from 0.0 to 0.6), with the best results typically observed around $\beta = 0.4$ or 0.6. Notably, performance at $\beta = 0$ using only relation-guided attention—is nearly as strong, suggesting that contextual semantics alone provide valuable guidance. However, performance declines gradually at $\beta = 0.8$ and drops sharply at $\beta = 1.0$, where only global tokens from the CLIP encoder are used without any relation-specific modulation. This highlights the importance of integrating both global and contextual signals for relational alignment.

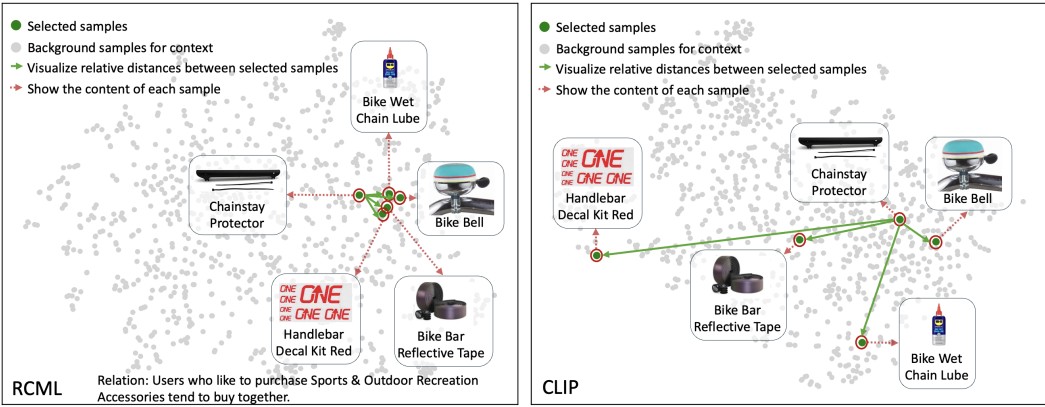

Figure 6: t-SNE visualization of multimodal embeddings under the semantic relation *"Co-purchased by people who primarily buy Sports & Outdoor Recreation Accessories."*.

**Visualization.** To qualitatively evaluate our framework, we visualize the feature spaces learned by RCML and CLIP using t-SNE. We randomly sample a subset of products and extract their embeddings under the relation "Users who like to purchase Sports & Outdoor Recreation Accessories tend to buy together." As shown in Figure 6, RCML produces more compact and semantically meaningful clusters. For example, items such as bike bells, chainstay protectors, and wet chain lubes are grouped closely together, reflecting their shared relevance to cycling enthusiasts. In contrast, CLIP embeddings appear more dispersed, indicating a lack of relation-aware organization. These results demonstrate that our model goes beyond surface-level similarity, capturing functional and intent-driven associations.

**Case Study.** Figure 7 shows representative examples from the Relation-Guided Retrieval task on Amazon dataset. In each case, the leftmost item (highlighted with a yellow dashed box) is the query $A$, and the top-3 retrieved candidates appear on the right.For convenience, each semantic relation is abbreviated as a label and displayed on the connecting arrow. Ground-truth targets are marked with

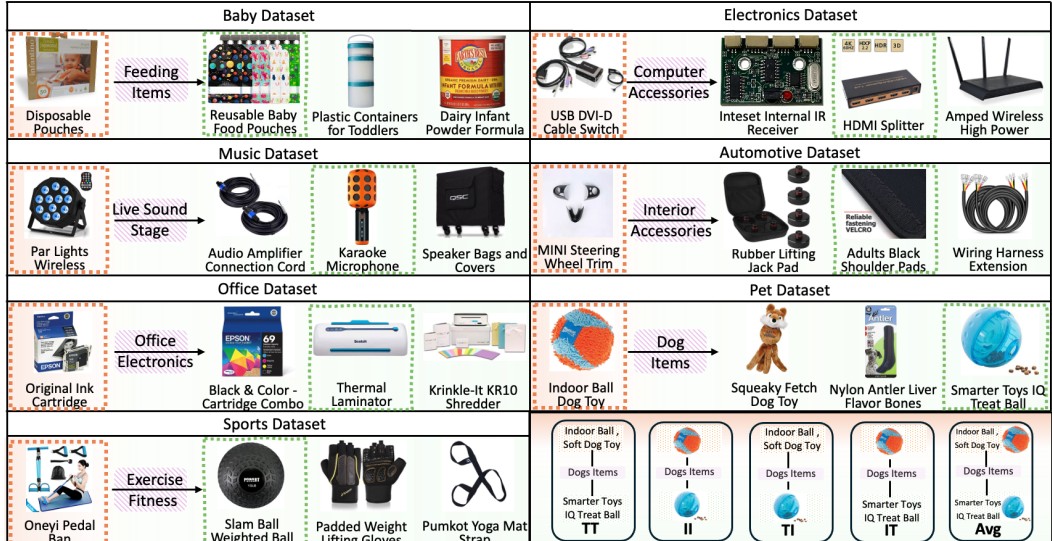

Figure 7: Case study showing the top-3 predictions ranked by our model in Relation-Guided Retrieval. Highlighted items correspond to the query, relation context, and correct targets.

green boxes. As shown, our model consistently retrieves contextually relevant items aligned with the intended relation. For instance, given a karaoke microphone and the relation "Live Sound & Stage," the model retrieves accessories such as speaker bags and audio connectors, rather than irrelevant items with visual or textual similarity. The bottom panel illustrates the five similarity configurations (TT, II, TI, IT, AVG) used in scoring, which provide complementary perspectives for retrieval.

**Efficiency and Model Size.** Table 2 compares inference latency and model size across all methods. RCML computes relation-conditioned embeddings at inference time, yet remains efficient (14.32 ms/sample, 152.33M parameters), only slightly above CLIP and DeCLIP. Notably, Image-Bind incurs much higher cost (35.90 ms/sample, 1200M+ parameters) while still underperforming RCML on most tasks, making it less practical for retrieval scenarios. These results show that relation conditioning introduces minimal overhead while delivering superior performance.

## 5 CONCLUSION

We presented RCML, a contrastive learning framework that conditions multimodal representation learning on semantic relations. Across multiple datasets and tasks, RCML consistently outperforms strong baselines. Further analysis shows that RCML delivers robust performance across domains and effectively organizes items under relation-defined contexts by bringing semantically related products closer in the embedding space. Beyond its empirical strength, RCML offers a general and adaptable learning paradigm that can be integrated into more advanced multimodal systems to support relation-aware representation learning.

Table 2: Inference time and parameter count for each model.

| Model | Inference Time (ms/sample) | #Params (M) |
|---|---|---|
| CLIP | 9.17 | 151.28 |
| DeCLIP | 11.29 | 158.76 |
| UniCL | 21.93 | 150.70 |
| SigLIP | 9.89 | 203.16 |
| ImageBind | 35.90 | 1200.78 |
| RCML | 14.32 | 152.33 |

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

# A APPENDIX

## A.1 IMPLEMENTATION FRAMEWORK.

Our RCML model and CLIP baseline are implemented using the Hugging Face CLIPModel and CLIPProcessor with a ViT-B/32 backbone. Other baselines (DeCLIP, UniCL, SigLIP, ImageBind) are evaluated using their released checkpoints. All reported results are averaged over three runs with different random seeds.

We perform grid search over multiple hyperparameters and select the best setting based on validation performance. The final configuration is as follows: batch size of 512, AdamW optimizer, learning rate of $5 \times 10^{-5}$ with cosine decay, and early stopping based on validation performance (training typically converged within 3 epochs). The contrastive temperature $\tau$ is set to 0.1, the intra-modal weight $\lambda$ to 0.5, and the attention balance coefficient $\beta$ to 0.6.

All experiments are conducted on a single NVIDIA RTX A6000 GPU with 48GB memory. The software environment consists of Python 3.9, PyTorch 1.12, Transformers 4.26, and CUDA 11.6.

## A.2 EDGE TEXT DESCRIPTIONS

For intra-sample relations, all datasets share the same edge text: *"These two items represent the same product. Align their features accordingly."*

For inter-sample relations, the relation texts are dataset-specific. To obtain relation texts $e_{ij}$, we follow established practices in recommender systems that use user clustering and profiling to capture group-level preferences (Zhang et al., 2016) (Adomavicius & Tuzhilin, 2005). Concretely, we represent each user's purchase or reading history as a distribution over product categories, cluster users based on these distributions, and summarize each cluster with a natural-language description that converts structured statistics into interpretable context. This design grounds edges in real user behavior while providing semantic descriptions that serve as contextual input for relation-aware representation learning. Detailed examples are provided below:

### A.2.1 DATASET: BABY

| Cluster | Edge Text Description |
|---------|----------------------|
| Cluster_0 | Co-purchased by people who primarily buy Car Seats & Accessories, along with Nursery and Feeding, while showing lower engagement in Diapering, Baby Care, and Safety, and minimal interest in Strollers & Accessories, Activity & Entertainment, Gifts, and Potty Training. |
| Cluster_1 | Co-purchased by people who overwhelmingly favor Feeding, with much lower interest in Nursery, Diapering, Baby Care, Safety, Strollers & Accessories, Car Seats & Accessories, Gifts, Activity & Entertainment, and Potty Training. |
| Cluster_2 | Co-purchased by people who strongly prefer Safety, while also buying Nursery and Feeding, with moderate interest in Diapering and Baby Care, and minimal interest in Strollers & Accessories, Car Seats & Accessories, Gifts, Activity & Entertainment, and Potty Training. |
| Cluster_3 | Co-purchased by people who heavily favor Diapering, with secondary but lower preference for Nursery and Feeding, minor engagement in Baby Care and Safety, and almost no interest in Strollers & Accessories, Car Seats & Accessories, Gifts, Activity & Entertainment, and Potty Training. |
| Cluster_4 | Co-purchased by people who strongly favor Potty Training, with balanced interest in Nursery, Feeding, and Diapering, while showing low purchases in Baby Care and Safety, and negligible interest in Strollers & Accessories, Car Seats & Accessories, Gifts, and Activity & Entertainment. |
| Cluster_5 | Co-purchased by people who dominantly prefer Baby Care, followed by Nursery and Feeding, while showing almost no interest in Diapering, Safety, Strollers & Accessories, Car Seats & Accessories, Gifts, Activity & Entertainment, and Potty Training. |

| Cluster | Edge Text Description |
|---|---|
| Cluster_6 | Co-purchased by people who are highly engaged in Activity & Entertainment, together with Nursery and Feeding, while showing relatively low purchases of Diapering, Baby Care, and Safety, and minimal interest in Strollers & Accessories, Car Seats & Accessories, Gifts, and Potty Training. |
| Cluster_7 | Co-purchased by people who overwhelmingly buy Gifts, along with Nursery and Feeding, while showing significantly lower interest in Diapering, Baby Care, Safety, Strollers & Accessories, Car Seats & Accessories, Activity & Entertainment, and Potty Training. |
| Cluster_8 | Co-purchased by people who show a striking preference for Strollers & Accessories, with secondary interest in Nursery and Feeding, while showing low purchases in Diapering, Baby Care, and Safety, and minimal interest in Car Seats & Accessories, Gifts, Activity & Entertainment, and Potty Training. |
| Cluster_9 | Co-purchased by people who primarily buy Nursery, followed by Feeding, Diapering, and Baby Care, while showing minimal interest in Safety, Car Seats & Accessories, Strollers & Accessories, Gifts, Activity & Entertainment, and Potty Training. |

### A.2.2 DATASET: ELECTRONICS

| Cluster | Edge Text Description |
|---|---|
| Cluster_0 | Co-purchased by people who primarily buy Computers & Accessories, along with Camera & Photo, while showing lower engagement in Television & Video, Headphones, Earbuds & Accessories, and Car & Vehicle Electronics, and minimal interest in Portable Audio & Video, Home Audio, Accessories & Supplies, Wearable Technology, and Power Accessories. |
| Cluster_1 | Co-purchased by people who overwhelmingly favor Wearable Technology, with much lower interest in Computers & Accessories, Camera & Photo, Television & Video, Headphones, Earbuds & Accessories, Car & Vehicle Electronics, Portable Audio & Video, Home Audio, Accessories & Supplies, and Power Accessories. |
| Cluster_2 | Co-purchased by people who strongly prefer Portable Audio & Video, while also buying Computers & Accessories and Camera & Photo, with moderate interest in Television & Video, Headphones, Earbuds & Accessories, and Car & Vehicle Electronics, and minimal interest in Home Audio, Accessories & Supplies, Wearable Technology, and Power Accessories. |
| Cluster_3 | Co-purchased by people who heavily favor Camera & Photo, with secondary but lower preference for Computers & Accessories, minor engagement in Television & Video, Headphones, Earbuds & Accessories, Car & Vehicle Electronics, and Portable Audio & Video, and almost no interest in Home Audio, Accessories & Supplies, Wearable Technology, and Power Accessories. |
| Cluster_4 | Co-purchased by people who strongly favor Car & Vehicle Electronics, with balanced interest in Computers & Accessories and Camera & Photo, while showing low purchases in Television & Video and Headphones, Earbuds & Accessories, and negligible interest in Portable Audio & Video, Home Audio, Accessories & Supplies, Wearable Technology, and Power Accessories. |

| Cluster | Edge Text Description |
|---------|---------------------|
| Cluster_5 | Co-purchased by people who dominantly prefer Television & Video, followed by Computers & Accessories and Camera & Photo, while showing almost no interest in Headphones, Earbuds & Accessories, Car & Vehicle Electronics, Portable Audio & Video, Home Audio, Accessories & Supplies, Wearable Technology, and Power Accessories. |
| Cluster_6 | Co-purchased by people who are highly engaged in Power Accessories, together with Computers & Accessories and Camera & Photo, while showing relatively low purchases of Television & Video, Headphones, Earbuds & Accessories, Car & Vehicle Electronics, Portable Audio & Video, Home Audio, Accessories & Supplies, and Wearable Technology. |
| Cluster_7 | Co-purchased by people who overwhelmingly buy Headphones, Earbuds & Accessories, along with Computers & Accessories and Camera & Photo, while showing significantly lower interest in Television & Video, Car & Vehicle Electronics, Portable Audio & Video, Home Audio, Accessories & Supplies, Wearable Technology, and Power Accessories. |
| Cluster_8 | Co-purchased by people who show a striking preference for Home Audio, with secondary interest in Computers & Accessories and Camera & Photo, while showing low purchases in Television & Video, Headphones, Earbuds & Accessories, Car & Vehicle Electronics, and Portable Audio & Video, and minimal interest in Accessories & Supplies, Wearable Technology, and Power Accessories. |
| Cluster_9 | Co-purchased by people who primarily buy Accessories & Supplies, followed by Computers & Accessories and Camera & Photo, while showing minimal interest in Television & Video, Headphones, Earbuds & Accessories, Car & Vehicle Electronics, Portable Audio & Video, Home Audio, Wearable Technology, and Power Accessories. |

### A.2.3 DATASET: MUSICAL INSTRUMENTS

| Cluster | Edge Text Description |
|---------|---------------------|
| Cluster_0 | Co-purchased by people who overwhelmingly buy Instrument Accessories, with significantly lower engagement in Live Sound & Stage, Microphones & Accessories, Drums & Percussion, and Guitars, and minimal interest in Studio Recording Equipment, Amplifiers & Effects, Electronic Music, DJ & Karaoke, Keyboards & MIDI, and Band & Orchestra. |
| Cluster_1 | Co-purchased by people who primarily buy Studio Recording Equipment, followed by Instrument Accessories, with moderate interest in Live Sound & Stage, Microphones & Accessories, Drums & Percussion, and Guitars, while showing minimal interest in Amplifiers & Effects, Keyboards & MIDI, Electronic Music, DJ & Karaoke, and Band & Orchestra. |
| Cluster_2 | Co-purchased by people who strongly prefer Electronic Music, DJ & Karaoke, while also buying Instrument Accessories, with moderate interest in Live Sound & Stage, Microphones & Accessories, Drums & Percussion, Guitars, and Studio Recording Equipment, and minimal engagement in Amplifiers & Effects, Keyboards & MIDI, and Band & Orchestra. |

| Cluster | Edge Text Description |
|---------|---------------------|
| Cluster_3 | Co-purchased by people who heavily favor Microphones & Accessories, while also purchasing Instrument Accessories, with moderate engagement in Live Sound & Stage and minimal interest in Drums & Percussion, Guitars, Studio Recording Equipment, Amplifiers & Effects, Electronic Music, DJ & Karaoke, Keyboards & MIDI, and Band & Orchestra. |
| Cluster_4 | Co-purchased by people who primarily buy Band & Orchestra, followed by Instrument Accessories, with moderate interest in Live Sound & Stage, Microphones & Accessories, Drums & Percussion, Guitars, and Studio Recording Equipment, while showing minimal engagement in Amplifiers & Effects, Electronic Music, DJ & Karaoke, and Keyboards & MIDI. |
| Cluster_5 | Co-purchased by people who heavily favor Drums & Percussion, followed by Instrument Accessories, with moderate interest in Live Sound & Stage and Microphones & Accessories, while showing minimal engagement in Studio Recording Equipment, Guitars, Amplifiers & Effects, Electronic Music, DJ & Karaoke, Keyboards & MIDI, and Band & Orchestra. |
| Cluster_6 | Co-purchased by people who strongly prefer Keyboards & MIDI, followed by Instrument Accessories, with moderate interest in Live Sound & Stage, Microphones & Accessories, Drums & Percussion, Guitars, and Studio Recording Equipment, while showing minimal engagement in Amplifiers & Effects, Electronic Music, DJ & Karaoke, and Band & Orchestra. |
| Cluster_7 | Co-purchased by people who overwhelmingly buy Live Sound & Stage, with significant engagement in Instrument Accessories, while showing minimal interest in Microphones & Accessories, Drums & Percussion, Guitars, Studio Recording Equipment, Amplifiers & Effects, Electronic Music, DJ & Karaoke, Keyboards & MIDI, and Band & Orchestra. |
| Cluster_8 | Co-purchased by people who primarily buy Amplifiers & Effects, followed by Instrument Accessories, with moderate engagement in Live Sound & Stage, Microphones & Accessories, Drums & Percussion, Guitars, and Studio Recording Equipment, while showing minimal interest in Keyboards & MIDI, Band & Orchestra, and Electronic Music, DJ & Karaoke. |
| Cluster_9 | Co-purchased by people who primarily buy Guitars, followed by Instrument Accessories, with moderate interest in Live Sound & Stage, Microphones & Accessories, and Drums & Percussion, while showing minimal engagement in Studio Recording Equipment, Amplifiers & Effects, Electronic Music, DJ & Karaoke, Keyboards & MIDI, and Band & Orchestra. |

### A.2.4 DATASET: AUTOMOTIVE

| Cluster | Edge Text Description |
|---------|---------------------|
| Cluster_0 | Co-purchased by people who overwhelmingly buy Replacement Parts, with significantly lower engagement in Motorcycle & Powersports, Exterior Accessories, and Interior Accessories, and minimal interest in Lights & Lighting Accessories, Tires & Wheels, Tools & Equipment, Car Care, RV Parts & Accessories, and Paint & Paint Supplies. |

| Cluster | Edge Text Description |
|---|---|
| Cluster_1 | Co-purchased by people who primarily buy Motorcycle & Powersports, followed by Replacement Parts, with moderate interest in Exterior Accessories and Interior Accessories, while showing minimal engagement in Lights & Lighting Accessories, Tires & Wheels, Tools & Equipment, Car Care, RV Parts & Accessories, and Paint & Paint Supplies. |
| Cluster_2 | Co-purchased by people who strongly prefer Lights & Lighting Accessories, followed by Replacement Parts, with moderate interest in Motorcycle & Powersports, Exterior Accessories, and Interior Accessories, while showing minimal engagement in Tires & Wheels, Tools & Equipment, Car Care, RV Parts & Accessories, and Paint & Paint Supplies. |
| Cluster_3 | Co-purchased by people who heavily favor Interior Accessories, followed by Replacement Parts, with moderate engagement in Motorcycle & Powersports and Exterior Accessories, while showing minimal interest in Lights & Lighting Accessories, Tires & Wheels, Tools & Equipment, Car Care, RV Parts & Accessories, and Paint & Paint Supplies. |
| Cluster_4 | Co-purchased by people who primarily buy Car Care, followed by Replacement Parts, with moderate interest in Motorcycle & Powersports, Exterior Accessories, and Interior Accessories, while showing minimal engagement in Lights & Lighting Accessories, Tires & Wheels, Tools & Equipment, RV Parts & Accessories, and Paint & Paint Supplies. |
| Cluster_5 | Co-purchased by people who heavily favor Exterior Accessories, followed by Replacement Parts, with moderate interest in Motorcycle & Powersports, while showing minimal engagement in Interior Accessories, Lights & Lighting Accessories, Tires & Wheels, Tools & Equipment, Car Care, RV Parts & Accessories, and Paint & Paint Supplies. |
| Cluster_6 | Co-purchased by people who strongly prefer RV Parts & Accessories, followed by Replacement Parts, with moderate interest in Motorcycle & Powersports, Exterior Accessories, and Interior Accessories, while showing minimal engagement in Lights & Lighting Accessories, Tires & Wheels, Tools & Equipment, Car Care, and Paint & Paint Supplies. |
| Cluster_7 | Co-purchased by people who primarily buy Paint & Paint Supplies, followed by Replacement Parts, with moderate interest in Motorcycle & Powersports, Exterior Accessories, and Interior Accessories, while showing minimal engagement in Lights & Lighting Accessories, Tires & Wheels, Tools & Equipment, Car Care, and RV Parts & Accessories. |
| Cluster_8 | Co-purchased by people who heavily favor Tires & Wheels, followed by Replacement Parts, with moderate engagement in Motorcycle & Powersports, Exterior Accessories, and Interior Accessories, while showing minimal interest in Lights & Lighting Accessories, Tools & Equipment, Car Care, RV Parts & Accessories, and Paint & Paint Supplies. |
| Cluster_9 | Co-purchased by people who primarily buy Tools & Equipment, followed by Replacement Parts, with moderate engagement in Motorcycle & Powersports, Exterior Accessories, and Interior Accessories, while showing minimal interest in Lights & Lighting Accessories, Tires & Wheels, Car Care, RV Parts & Accessories, and Paint & Paint Supplies. |

### A.2.5 DATASET: OFFICE PRODUCTS

| Cluster | Edge Text Description |
|---|---|
| Cluster_0 | Co-purchased by people who overwhelmingly buy Office & School Supplies, with much lower engagement in Office Electronics and Office Furniture & Lighting, and minimal interest in Education Store, Brother Remf Ink & Toner, Office Organization, Brands, Office Supplies Outlet, Leather Bags, and promotional discounts. |
| Cluster_1 | Co-purchased by people who primarily buy Office Electronics, followed closely by Office & School Supplies, with moderate engagement in Office Furniture & Lighting, while showing minimal interest in Education Store, Brother Remf Ink & Toner, Office Organization, Brands, Office Supplies Outlet, Leather Bags, and promotional discounts. |
| Cluster_2 | Co-purchased by people who strongly prefer promotional discounts (e.g., Elmers, Sharpie), followed by Office & School Supplies, with moderate interest in Office Electronics and Office Furniture & Lighting, while showing no engagement in Education Store, Office Supplies Outlet, Brands, Office Organization, Brother Remf Ink & Toner, or Leather Bags. |
| Cluster_3 | Co-purchased by people who heavily favor Office Furniture & Lighting, followed by Office & School Supplies, with moderate engagement in Office Electronics, while showing minimal interest in Education Store, Brother Remf Ink & Toner, Office Organization, Office Supplies Outlet, Brands, Leather Bags, and promotional discounts. |
| Cluster_4 | Co-purchased by people who primarily buy Brother Remf Ink & Toner, followed by Office & School Supplies, with moderate interest in Office Electronics and Office Furniture & Lighting, while showing minimal engagement in Education Store, Office Supplies Outlet, Brands, Office Organization, promotional discounts, and Leather Bags. |
| Cluster_5 | Co-purchased by people who heavily favor Leather Bags, followed by Office & School Supplies, with moderate interest in Office Electronics and Office Furniture & Lighting, while showing minimal engagement in Education Store, Brands, Office Supplies Outlet, Office Organization, Brother Remf Ink & Toner, and promotional discounts. |
| Cluster_6 | Co-purchased by people who strongly prefer Office Supplies Outlet, followed by Office & School Supplies, with moderate engagement in Office Electronics and Office Furniture & Lighting, while showing minimal interest in Education Store, Brands, Office Organization, Brother Remf Ink & Toner, promotional discounts, and Leather Bags. |
| Cluster_7 | Co-purchased by people who primarily buy Office & School Supplies and Brands, with moderate engagement in Office Electronics and Office Furniture & Lighting, while showing minimal interest in Education Store, Office Supplies Outlet, Office Organization, Brother Remf Ink & Toner, promotional discounts, and Leather Bags. |
| Cluster_8 | Co-purchased by people who heavily favor Office & School Supplies and Office Organization, with moderate engagement in Office Electronics and Office Furniture & Lighting, while showing minimal interest in Education Store, Office Supplies Outlet, Brands, Brother Remf Ink & Toner, promotional discounts, and Leather Bags. |

| Cluster | Edge Text Description |
|---------|---------------------|
| Cluster_9 | Co-purchased by people who primarily buy from the Education Store, followed by Office & School Supplies, with moderate engagement in Office Electronics and Office Furniture & Lighting, while showing no interest in promotional discounts, Brother Remf Ink & Toner, Office Supplies Outlet, Brands, Office Organization, or Leather Bags. |

### A.2.6  DATASET: PET SUPPLIES

| Cluster | Edge Text Description |
|---------|---------------------|
| Cluster_0 | Co-purchased by people who primarily buy Cats and Dogs, with moderate engagement in Fish & Aquatic Pets and Birds, while showing minimal interest in Small Animals, Horses, Reptiles & Amphibians, Top Dog Supplies, Top Cat Supplies, and Top Selection from AmazonPets. |
| Cluster_1 | Co-purchased by people who overwhelmingly buy Dogs, with significantly lower engagement in Cats and Fish & Aquatic Pets, while showing minimal interest in Birds, Small Animals, Horses, Reptiles & Amphibians, Top Dog Supplies, Top Cat Supplies, and Top Selection from AmazonPets. |
| Cluster_2 | Co-purchased by people who heavily favor Birds, followed by Dogs and Cats, with moderate engagement in Fish & Aquatic Pets, while showing minimal interest in Small Animals, Horses, Reptiles & Amphibians, Top Dog Supplies, Top Cat Supplies, and Top Selection from AmazonPets. |
| Cluster_3 | Co-purchased by people who primarily buy Fish & Aquatic Pets and Dogs, with moderate engagement in Cats, while showing minimal interest in Birds, Small Animals, Horses, Reptiles & Amphibians, Top Dog Supplies, Top Cat Supplies, and Top Selection from AmazonPets. |
| Cluster_4 | Co-purchased by people who heavily favor Horses, followed by Dogs and Cats, with moderate engagement in Fish & Aquatic Pets and Birds, while showing minimal interest in Small Animals, Reptiles & Amphibians, Top Dog Supplies, Top Cat Supplies, and Top Selection from AmazonPets. |
| Cluster_5 | Co-purchased by people who primarily buy Top Selection from AmazonPets, followed by Dogs and Cats, with moderate engagement in Fish & Aquatic Pets and Birds, while showing minimal interest in Small Animals, Horses, Reptiles & Amphibians, Top Dog Supplies, and Top Cat Supplies. |
| Cluster_6 | Co-purchased by people who heavily favor Small Animals, followed by Dogs and Cats, with moderate engagement in Fish & Aquatic Pets and Birds, while showing minimal interest in Horses, Reptiles & Amphibians, Top Dog Supplies, Top Cat Supplies, and Top Selection from AmazonPets. |
| Cluster_7 | Co-purchased by people who primarily buy Reptiles & Amphibians, followed by Dogs and Cats, with moderate engagement in Fish & Aquatic Pets and Birds, while showing minimal interest in Small Animals, Horses, Top Dog Supplies, Top Cat Supplies, and Top Selection from AmazonPets. |

| Cluster | Edge Text Description |
|---------|---------------------|
| Cluster_8 | Co-purchased by people who heavily favor Top Cat Supplies, followed by Dogs and Cats, with moderate engagement in Fish & Aquatic Pets and Birds, while showing minimal interest in Small Animals, Horses, Reptiles & Amphibians, Top Dog Supplies, and Top Selection from AmazonPets. |
| Cluster_9 | Co-purchased by people who primarily buy Top Dog Supplies, followed by Dogs and Cats, with moderate engagement in Fish & Aquatic Pets and Birds, while showing minimal interest in Small Animals, Horses, Reptiles & Amphibians, Top Selection from AmazonPets, and Top Cat Supplies. |

### A.2.7 DATASET: SPORTS

| Cluster | Edge Text Description |
|---------|---------------------|
| Cluster_0 | Co-purchased by people who primarily buy Sports & Outdoor Recreation Accessories, along with Sports, Exercise & Fitness, and Outdoor Recreation, while showing lower engagement in Fan Shop and Hunting & Fishing, and minimal interest in Clothing, Sports Medicine, Memorabilia Display & Storage, and Tennis & Racket. |
| Cluster_1 | Co-purchased by people who overwhelmingly favor Sports, with significantly lower interest in Outdoor Recreation, Sports & Outdoor Recreation Accessories, Exercise & Fitness, Fan Shop, Hunting & Fishing, Clothing, Sports Medicine, Memorabilia Display & Storage, and Tennis & Racket. |
| Cluster_2 | Co-purchased by people who strongly prefer Tennis & Racket, while also buying Sports, Outdoor Recreation, Exercise & Fitness, Fan Shop, and Sports & Outdoor Recreation Accessories, with moderate interest in Hunting & Fishing and Clothing, and minimal interest in Sports Medicine and Memorabilia Display & Storage. |
| Cluster_3 | Co-purchased by people who heavily favor Outdoor Recreation, with secondary but lower preference for Sports, and minor engagement in Sports & Outdoor Recreation Accessories, Fan Shop, Exercise & Fitness, and Hunting & Fishing, while showing almost no interest in Clothing, Sports Medicine, Memorabilia Display & Storage, and Tennis & Racket. |
| Cluster_4 | Co-purchased by people who strongly favor Clothing, with balanced interest in Outdoor Recreation, Exercise & Fitness, Sports, Fan Shop, and Sports & Outdoor Recreation Accessories, while showing low purchases in Hunting & Fishing and negligible interest in Sports Medicine, Memorabilia Display & Storage, and Tennis & Racket. |
| Cluster_5 | Co-purchased by people who dominantly prefer Fan Shop, followed by Sports, Outdoor Recreation, Exercise & Fitness, and Hunting & Fishing, while showing almost no interest in Sports & Outdoor Recreation Accessories, Clothing, Sports Medicine, Memorabilia Display & Storage, and Tennis & Racket. |
| Cluster_6 | Co-purchased by people who are highly engaged in Memorabilia Display & Storage, together with Sports, Fan Shop, Outdoor Recreation, Exercise & Fitness, Hunting & Fishing, and Sports & Outdoor Recreation Accessories, while showing relatively low purchases of Clothing, Sports Medicine, and no interest in Tennis & Racket. |

| Cluster | Edge Text Description |
|---|---|
| Cluster_7 | Co-purchased by people who overwhelmingly buy Hunting & Fishing, along with Outdoor Recreation, Sports, and Exercise & Fitness, while showing significantly lower interest in Sports & Outdoor Recreation Accessories, Fan Shop, Clothing, Sports Medicine, Memorabilia Display & Storage, and Tennis & Racket. |
| Cluster_8 | Co-purchased by people who show a striking preference for Sports Medicine, with secondary interest in Sports, Outdoor Recreation, Exercise & Fitness, Fan Shop, Sports & Outdoor Recreation Accessories, and Hunting & Fishing, while showing minimal interest in Clothing, and no purchases in Memorabilia Display & Storage or Tennis & Racket. |
| Cluster_9 | Co-purchased by people who primarily buy Exercise & Fitness, followed by Sports, Outdoor Recreation, Sports & Outdoor Recreation Accessories, and Fan Shop, while showing minimal interest in Hunting & Fishing, Clothing, Sports Medicine, Memorabilia Display & Storage, and Tennis & Racket. |

### A.2.8 DATASET: GOODREADS

| Cluster | Edge Text Description |
|---|---|
| Cluster_0 | Co-read mostly by people whose primary interest is Children's literature, while Fantasy & Paranormal frequently appears as a secondary theme. |
| Cluster_1 | Co-read mainly by readers drawn to Comics & Graphic works, with History & Biography often forming a complementary interest. |
| Cluster_2 | Frequently co-read by those immersed in Fantasy & Paranormal, who also tend to branch into Mystery, Thriller & Crime. |
| Cluster_3 | Predominantly co-read by readers focused on History & Biography, who also show a marked tendency toward Mystery, Thriller & Crime. |
| Cluster_4 | Typically co-read by people with a strong taste for Mystery, Thriller & Crime, while Romance emerges as a notable accompanying category. |
| Cluster_5 | Commonly co-read by readers who appreciate Poetry, with History & Biography serving as a frequent additional interest. |
| Cluster_6 | Co-read largely by readers whose core preference lies in Romance, and who often extend their engagement into Young Adult literature. |
| Cluster_7 | Co-read primarily by readers centered on Young Adult, with Romance frequently appearing as a closely associated genre. |

