# OpenReview forum: "Multimodal Representation Learning Conditioned on Semantic Relations"
_ICLR.cc/2026/Conference — Submitted to ICLR 2026_

### Official Review · Reviewer_SLmC · 2025-10-24

**Soundness:** 2
**Presentation:** 2
**Contribution:** 2
**Rating:** 4
**Confidence:** 3

**Summary:**

This paper proposes a new multimodal representation learning framework that conditions on the semantic relations between different samples. Beyond leveraging paired text–image data, it further defines and explores inter-sample relationships and utilizes these connections to guide feature extraction and alignment. The framework is optimized using contrastive losses among image–text, text–image, text–text, and image–image pairs to enhance both cross-modal and intra-modal consistency.

**Strengths:**

- It is interesting to exploit unpaired data by modeling potential semantic relations between samples.

- The proposed method is evaluated on three downstream tasks, namely the relation-guided retrieval, relation type prediction, and relation validity prediction, where the results demonstrate performance gains over existing multimodal representation learning approaches.

**Weaknesses:**

- The proposed framework appears specifically tailored for recommendation systems. However, in the evaluation, it is compared mainly against general-purpose multimodal representation learning methods, which may not be a fair comparison. The authors should also consider including baselines from existing recommendation approaches.

- The paper misses several important related works in multimodal representation learning, such as VAST: Vision-Audio-Subtitle-Text Omni-Modality Foundation Model and Dataset (NeurIPS 2023), as well as relevant discussions on existing recommendation systems.

- The presentation quality needs improvement. The current version contains typographical errors, missing spaces, and incomplete references (e.g., “?”). The authors should carefully proofread and polish the manuscript.

**Questions:**

Can the proposed method be generalized and applied to tasks beyond recommendation systems?

---

> ### Author Response · Authors · 2025-11-28
>
> We thank the reviewer for the thoughtful comments.
>
> **W1- Q1** To clarify, RCML is not a recommendation model, but a general-purpose multimodal representation learning framework that introduces explicit semantic relation conditioning using free-form natural-language descriptions. This capability is fundamentally different from recommendation systems, which rely on user–item interactions, collaborative filtering structures, or task-specific objectives.
>
> A key advantage of RCML is that it produces relation-conditioned representations, allowing the same item to obtain different embeddings under different semantic contexts. Such context-dependent modulation is not supported by recommendation models nor by standard multimodal contrastive learning, whose embeddings remain fixed and relation-agnostic. RCML therefore addresses a representation-learning challenge—how to integrate semantic relations directly into the feature space—rather than optimizing for recommendation behavior.
>
> We use Amazon and Goodreads datasets as they provide rich many-to-many semantic relations necessary for evaluating relation-conditioned learning. However, the framework itself is domain-agnostic, and can be applied to multimodal document modeling, news propagation chains, or any domain where semantic relations exist. We will revise the introduction to make this generality explicit.
>
> **W2-W3** Regarding related works, we appreciate the reviewer’s pointers. We will incorporate a brief discussion of multimodal foundation models such as VAST (NeurIPS 2023) and clarify why these models, while powerful, do not address semantic relation conditioning. We will also add a short note on conventional recommendation approaches to explain why they are not directly comparable. Finally, we will carefully proofread and correct all formatting and citation inconsistencies in the revised version.

---

> > ### Comment · Area_Chair_NaC7 · 2025-11-28
> >
> > Dear Reviewer,
> >
> > Please make sure you read the authors' response and engage with them in the discussion before the end of the discussion period on **Dec 03 '25 09:00 PM UTC**. This is a hard deadline.
> >
> > Thank you for supporting quality peer review at ICLR.
> >
> > AC

---

### Official Review · Reviewer_ykbu · 2025-10-27

**Soundness:** 2
**Presentation:** 2
**Contribution:** 3
**Rating:** 4
**Confidence:** 3

**Summary:**

This paper introduces Relation-Conditioned Multimodal Learning (RCML), a framework designed to enhance multimodal representation learning by explicitly incorporating semantic relations between items. The core idea is to move beyond the simple pairwise alignment of image-text pairs (as in CLIP) and instead model many-to-many relationships described by natural language. The method achieves this through a novel relation-guided attention mechanism, where the embedding of a relation's text description modulates the feature extraction from image and text modalities.

**Strengths:**

1. The primary contribution, using natural language descriptions of relations to directly condition the representation learning process, is to the best of my knowledge novel and addresses a gap in the current vision-language model.

2. RCML consistently and significantly outperforms strong baselines, including CLIP, SigLIP, and ImageBind, on several well-designed tasks.

3. The paper evaluates the model on three distinct tasks (retrieval, type prediction, validity prediction) that effectively test the claimed capabilities. The ablation studies (Fig. 4) clearly demonstrate the importance of each proposed component, especially the inter-sample relations and their semantic descriptions.

**Weaknesses:**

1. In the main retrieval task (4.3.1), the strategy for providing relational context to baseline models (i.e., concatenating the relation text with the item description) may not be the strongest form of comparison. It is unclear if this simple concatenation allows models like CLIP to effectively leverage the relational information. A more rigorous baseline would involve fully fine-tuning a model like CLIP on this task, which would better isolate the architectural benefits of RCML's explicit modulation mechanism.

2. The relation type prediction task (4.3.2) demonstrates RCML's capabilities but lacks a comparison against any baseline. A simple baseline, such as a lightweight classifier trained on top of concatenated frozen features from a standard CLIP model, would help interpreting RCML's performance and establish the difficulty of the task.

3. While the paper mentions the limitations of pairwise methods, the introduction (lines 38-51) could be strengthened by citing relevant work on relational learning from other fields (e.g., knowledge graph-based recommendation) to better situate the problem in the broader literature. The paper also contains several missing citations (marked with "?") in the related work section, that suggests writing still needs some polishing.

4. There is growing evidence that alignment emerges between latent spaces without explicit optimization to enforce it (few examples: [1,2]) . I think it would be beneficial to briefly discuss how this work positions with respect to those, especially for [2] that explicitly makes use of relations between instances.

---

[1] Huh,  et al, "The Platonic Representation Hypothesis," 2024.

[2] Moschella, et al, "Relative representations enable zero-shot latent space communication," in ICLR, 2023.

**Questions:**

1. The framework uses the same CLIP text encoder for both item descriptions and the natural language relation descriptions. This implies that these two types of text live in the same embedding space. Could the authors elaborate on this design choice? Have the authors considered whether this is optimal, or if using separate projectors or encoders for relations might offer benefits?

2. An ablation to show that relations cannot learned implicitly from the data would improve the manuscript.
Would an experiment where a standard CLIP model is fully fine-tuned on the relation-guided retrieval task be possible? This would help clarify whether the performance gains come from the proposed architecture or simply from training/fine-tuning on relational data.

3. In the relation-guided retrieval task (4.3.1), the manuscript report results on "36 out of 40 metrics.". If my understanding is correct, these should be 5 similarity scores across 8 datasets.

---

> ### Author Response · Authors · 2025-11-28
>
> Thank you for the reviewer’s constructive feedback. We address each point below.
>
> **W1–Q2.** Our evaluation in Section 4.3.1 is designed to compare how pretrained multimodal encoders respond to relational context without introducing task-specific finetuning, which would change the learning paradigm and obscure the architectural contribution of RCML. Fully fine-tuning CLIP (or SigLIP / ImageBind) would shift the comparison toward supervised training scale rather than isolating the effect of relation conditioning, and such models still cannot produce context-dependent embeddings controlled by semantic relation text. For baseline models that cannot make use of relation semantics, concatenation provides a practical and architecture-agnostic way to expose relation text during evaluation without modifying their training objectives. We will clarify this evaluation strategy in the revision.
>
> **W2.** We appreciate the reviewer’s suggestion. In our current design, the relation type prediction task in Section 4.3.2 is intended as a zero-shot probe of how much relation semantics are encoded in RCML’s relation-conditioned representations, rather than as a general benchmark for arbitrary encoders. Standard models such as CLIP, SigLIP, or ImageBind do not produce relation-conditioned embeddings and therefore cannot perform this task without adding a trained classifier or fine-tuning on relation labels. Introducing such supervision would change the nature of the task and make it closer in spirit to our supervised experiment (4.3.3). We will clarify this positioning in the revision.
>
> **W3–W4.** We will expand the related-work section to more clearly situate our approach. Prior work such as PRH [1] and Relative Representations [2] investigates global latent-space alignment or communication across representation spaces, rather than how embeddings should adapt under specific semantic relations. RCML is complementary to these directions: it leverages general alignment phenomena but addresses a distinct objective—learning relation-sensitive, context-dependent multimodal representations guided by natural-language relational descriptions. The revision will clarify these distinctions and incorporate the relevant citations.
>
> **Q1.**  In our setting, both item descriptions and relation descriptions are natural-language text, and placing them in the same semantic space is important for effective conditioning. Using a shared text encoder follows standard practice in multimodal alignment, where all textual inputs share the same representation space to maintain compatibility with the image encoder. Using a completely separate encoder for relation text would place items and relations in disconnected embedding spaces and make relation-conditioned attention substantially less stable. For this reason, we adopt a shared encoder, which is the conventional and empirically reliable design. We will clarify this rationale in the revision.
>
> **Q3.** Thank you for pointing this out. The “36 out of 40 metrics” corresponds to 5 similarity metrics evaluated across 8 datasets. We will correct the phrasing for clarity.
>
> [1] Huh, M., Cheung, B., Wang, T. and Isola, P., 2024. The platonic representation hypothesis. arXiv preprint arXiv:2405.07987.
> [2] Moschella, L., Maiorca, V., Fumero, M., Norelli, A., Locatello, F. and Rodolà, E., 2023, January. Relative representations enable zero-shot latent space communication. In ICLR.

---

> > ### Comment · Area_Chair_NaC7 · 2025-11-28
> >
> > Dear Reviewer,
> >
> > Please make sure you read the authors' response and engage with them in the discussion before the end of the discussion period on **Dec 03 '25 09:00 PM UTC**. This is a hard deadline.
> >
> > Thank you for supporting quality peer review at ICLR.
> >
> > AC

---

> > ### Comment · Reviewer_ykbu · 2025-11-28
> >
> > Thank you for your detailed response. I appreciate the planned clarifications regarding the shared text encoder and the related works.
> >
> > However, my primary concern regarding the experimental baselines remains. I understand the desire to evaluate zero-shot conditioning without fine-tuning. My point is that this setup makes it difficult to disentangle the gains from the proposed architecture versus the gains that could be achieved by simply exposing a standard architecture to relational data via fine-tuning.
> >
> > A fine-tuned baseline would serve as a crucial ablation to demonstrate that the architectural design of RCML, and not just the relational training data, is the primary driver of the performance gains.
> >
> > Due to this key reservation, my overall assessment remains unchanged. I am happy to discuss this further if I have misunderstood your rationale.

---

### Official Review · Reviewer_bTXS · 2025-10-31

**Soundness:** 2
**Presentation:** 2
**Contribution:** 1
**Rating:** 2
**Confidence:** 3

**Summary:**

This paper introduces relation-conditioned multimoidal learning, that incorporates relationships between samples for CLIP like encoder models. Specifically, the authors make use of inter-sample relationship by proposing a relation-guided attention mechanism as well as a hybridf loss function. Experiments showed improvements on classification and retrieval tasks.

**Strengths:**

- the paper is somewhat easy to read for which the reviewer appreciates!

**Weaknesses:**

- there has been a lot of work that tries to add prior knowledge of relationships between samples in the contrastive learning world. lots of different ways people have tried contrast samples with. in general not a fan of these line of work and not sure how much this paper contributes to that body of work.
- the baselines compared are old (<=2023) and not even sure how it's a valid comparison given their parameter count, training dataset size, etc. are all different.
- natural language relationships are not naturally occuring, they are generated suing clustering it seems (looking at Appendix) - correct me if i'm wrong. the authors might want to be clear about that these relationship does not come for free or naturally labeled in the dataset. the authors might also want to provide some analysis on how well this step introduces noisy relationships that might not help with learning.
- the evaluation is mostly on product oriented datasets as opposed to more general benchmarks. I would recommend authors benchmark its method on a few more datasets that is more broadly applicable.
- the authors should also compare with baselines that make use of semantic relationships.

**Questions:**

n/a

---

> ### Author Response · Authors · 2025-11-28
>
> Thank you for the suggestion.
>
> **W1.** Existing works that incorporate relationships into contrastive learning generally use relations only as auxiliary supervision—e.g., treating graph neighbors as positives, adjusting sampling strategies based on labels or metadata, or enforcing handcrafted relational constraints such as co-occurrence or temporal order. These approaches assume a *single* embedding per item and use relations merely to shape contrastive pairs.
>
> RCML is fundamentally different in both goal and mechanism. Instead of using relations to select or weight positives, RCML uses **free-form natural-language semantics of a relation to modulate the representation itself**, producing *different embeddings for the same item under different relational contexts*. To our knowledge, no prior contrastive or multimodal work supports this form of **relation-conditioned representation**. This capability is essential for many-to-many relational reasoning but is absent from existing methods.We will expand the related-work discussion to highlight this distinction clearly.
>
> **W2.** The baselines in our experiments—CLIP, DeCLIP, UniCL, SigLIP, and ImageBind—remain standard and widely adopted multimodal contrastive backbones. Differences in parameter count or pretraining data are inherent to the CLIP family and are routinely present in prior evaluations based on pretrained encoders. Our aim is not to compare pretraining regimes, but to test whether relation conditioning yields consistent improvements across strong and commonly used foundations. We will clarify this motivation in the revision.
>
> **W3.**  We confirm that the natural-language relation descriptions are automatically derived from clustering users based on their category-level purchase preference distributions. This is an intentional design choice: real-world multimodal datasets rarely provide explicit relation text, and RCML is meant to operate on naturally occurring relational structure rather than curated annotations. We will state this clearly in the revision.
>
> Regarding potential noise, we performed a lightweight sanity check by examining the consistency of user-preference patterns within each cluster. The dominant category preferences (e.g., top-3 favored categories and their relative ordering) remain stable across users in the same cluster, indicating that the automatically generated relation descriptions are semantically coherent even though they are not perfectly clean. These relations therefore provide sufficiently meaningful context for relation-conditioned learning. We will clarify this in the revision.
>
> **W4–W5.** Our evaluation sets are constructed from Amazon and Goodreads, which are standard large-scale multimodal corpora. They naturally contain interaction signals from which item–item relations can be derived, and we simply use these signals to build relational evaluation data. We will clarify this construction in the revision.
>
> For baselines, we incorporate the same relation information by concatenating the relation text with the item description, ensuring that all models receive identical relational cues. However, standard multimodal encoders still produce a single, relation-agnostic embedding for each item, whereas RCML uses the relation as an explicit conditioning signal and yields distinct embeddings under different relations. We will clarify this distinction.

---

> > ### Comment · Area_Chair_NaC7 · 2025-11-28
> >
> > Dear Reviewer,
> >
> > Please make sure you read the authors' response and engage with them in the discussion before the end of the discussion period on **Dec 03 '25 09:00 PM UTC**. This is a hard deadline.
> >
> > Thank you for supporting quality peer review at ICLR.
> >
> > AC

---

### Official Review · Reviewer_MwnY · 2025-11-01

**Soundness:** 1
**Presentation:** 2
**Contribution:** 2
**Rating:** 2
**Confidence:** 4

**Summary:**

The paper proposes a learnable pooling layer that can be plugged into any modality specific branch of a CLIP like model. The idea is to make this pooling conditional on a relation encoding, a vector encoding of a text description of what the aggregation should focus on. This allows the model to produce modality specific embeddings that can be compared under a given relation instead of relying on generic CLIP similarities.
Experiments on two datasets show promising results, and there are ablations for the main hyperparameters involved.

**Strengths:**

- The main idea of relation-conditioned representation learning is interesting and addresses a real limitation of standard CLIP objectives.
- Section 4.4 (sensitivity analysis) is a solid ablation. It explores the effect of β and provides insight into how the conditioning mechanism behaves.
- Using frozen CLIP backbones (ViT-B/32) keeps the setup clean and isolates the proposed contribution.
- The results are consistent across datasets.
- The code is attached and it looks structured and replicable.
- Figure 3 is beautiful!

**Weaknesses:**

### Clarity and terminology
- **Abstract (013–015)**: the expression “semantic relations across each pairs” is too vague. Even after reading the introduction, it’s not clear what kinds of relations are being modeled. If the goal is to keep the abstract general, it would still help to include one or two examples, or clarify this in the introduction. This is the central concept of the paper and should be grounded with concrete cases from the start.
- **Introduction (036–037)**: the list of “several directions” should include references (at least one per direction). The same applies to line 050 for the graph-based approaches.

### References and formatting
- Some references appear broken or missing (e.g., 095–096, 117–118). It might be useful to search for “?” in the PDF to find them.

### Method and conceptual clarity
- **Section 3 (last paragraph of 3.2)**: it seems there are no relation-specific negatives. If that’s correct, it’s a bit counterintuitive, since it would mean the model only learns to make items with *any* relation closer, regardless of which relation it is. Clarify whether that is the intention or if some relation-level contrast is used.
- **Section 3.3**: this part is difficult to follow. Please confirm if this interpretation is correct:
  > The implementation adds a modality-specific modulating layer on top of each CLIP branch. This layer performs a pooling step conditioned on a property encoded by the text branch. The resulting embedding is then interpolated with the standard CLS/EOT embedding, controlled by β.
  If this matches the authors’ design, it would help to rewrite this section more clearly along those lines.

### Experiments and reporting
- **Section 4.1 (265–266)**: “each relation is annotated with a natural-language description indicating its semantic context.” Since the paper previously treats the relation as an abstract concept, it would help to give a couple of concrete examples and mention that these descriptions come directly from the dataset.
- **Section 4.2 (last sentence)**: the claim “with significantly more parameters than ours” should include an approximate quantification and a pointer to Table 2.
- **Section 4.3.2 (end of page 6)**: “We report Top-3 Accuracy on all datasets where relation types are sufficiently meaningful to support evaluation.” Specify which datasets this applies to and how, so readers know what “sufficiently meaningful” means here.
- **Section 4.3.3 (supervised relation validation)**: the experiment lacks detail. It’s not clear how many samples the classifiers are trained on, what the ratio of positive to negative examples is, or what training strategy is used.
- **Section 4.4**: the sensitivity analysis is valuable, but it would be even stronger if paired with an analysis of how much of the original CLIP performance is retained. That would show the trade-off between relation specialization and general-purpose ability.

### Table 1 and metrics
- Consider adding the **average rank (or MRR)**. Hit@5 only tells whether the correct item appears in the top 5, not its exact position. Average rank provides finer resolution. The code already computes MRR, so this should be easy to add, for example in the appendix as a mirrored table.
- Clarify **how many samples** are used to compute the metrics. Section 4.3.1 mentions one positive and twenty negatives per query, but the total number of queries per domain is not reported.
- Report **standard deviation** across runs. Appendix A.1 mentions three seeds but no variance. Reporting mean ± std would make the results more interpretable in terms of stability and significance.

### Minor

- **Introduction (031)**: “compatible” would fit better than “unified”.
- Formatting issues occur in several citations (e.g., 106–107, 118, 291–292), including parentheses and spacing. The [`cleveref`](https://ctan.org/pkg/cleveref?lang=en) package could simplify this.
- There are small inconsistencies in punctuation spacing that should be corrected.
- **Section 3 (137)**: the statement “unlike traditional contrastive learning that operates only on matched pairs” is not fully accurate. Standard contrastive methods also use negative pairs, not only the diagonal of the similarity or distance matrix.

**Questions:**

1. Regarding Section 3 (last paragraph of 3.2): are there relation-specific negatives, or is the objective indeed relation-agnostic once a connection exists?
2. On Section 3.3: can you confirm whether the interpretation provided under "Method and conceptual clarity" matches your implementation?
3. For Table 1: how many total test samples per domain are used to compute the metrics, and what's the MRR?
4. For Section 4.3.3: how many samples were used to train and evaluate the binary classifiers, and what was the positive/negative ratio?
5. For Section 4.1: can you provide examples of the “natural-language descriptions” mentioned? I'm assuming these are directly from the datasets, is it correct?
7. For Section 4.4: how much of the original CLIP performance is preserved after relation conditioning?
8. Can you confirm that the CLIP baseline uses the OpenAI ViT-B/32 weights and not OpenCLIP or LAION implementations?

### Different CLIP baseline

The current CLIP baseline shows that standard CLIP embeddings are too generic to compare product encodings under a specific relation. That’s fine and expected: there’s nothing in CLIP that makes it focus on the relation itself. So I see it more as a baseline justifying the need of **conditioning on the relation**, in support of the research question of this paper.

A fairer baseline to compare against, still with no training, would be to compare the product encodings `zA` and `zB` (whatever their modality) **under** the similarity they each have with respect to the relation encoding `zr`. This gives a value in `[-1, 1]`,  the product of two cosine similarities, that reflects how much the two samples align with the relation.

```python
def relation_similarity(zA, zB, zr):
    zA, zB, zr = [x / x.norm() for x in (zA, zB, zr)]
    return (zA @ zr) * (zB @ zr)
```

And it can be used for both ranking (as it is) or to tune a threshold for classification purposes.

---

> ### Author Response · Authors · 2025-11-28
>
> We thank the reviewer for the very careful and thoughtful feedback. The comments are detailed and constructive, and we genuinely appreciate the effort. We address the points below and will incorporate the suggested clarifications in the revision.
>
> **Q1/Method and conceptual clarity.** There are no relation-specific negatives across different relations. This is intentional. For each relation r, the model contrasts its positive pairs against non-connected pairs *within that same relation context*, but we do not enforce that representations under different relations be pushed apart. RCML’s goal is to learn relation-conditioned embeddings—i.e., to let the same item take on different representations depending on the semantics of r—rather than to make different relations mutually exclusive in the embedding space. We will clarify this design choice in Section 3.2.
>
> **Q2 / Method and conceptual clarity.** Thank you for the careful reading. The interpretation is partially aligned with our design but differs in two key aspects. Our method does **not** introduce a modality-specific pooling layer, nor does it **interpolate CLS/EOT embeddings** with relation features. Instead:
>
> - The **relation description is encoded and used as the query** in a cross-attention module.
> - The item’s **text/image embeddings serve as key and value**.
> - The attention logits combine the relation query with a **binary token-selection mask** (controlled by β). This mask is introduced to **stabilize training by softly constraining attention toward semantically meaningful tokens** (e.g., EOT), while still allowing relation-driven reweighting. (An empirical analysis of β is provided in Fig. 5.)
> - The output of this cross-attention layer is the **relation-conditioned representation**, which is used directly.
>
> We will revise Section 3.3 to make this computation explicit and avoid the source of ambiguity.
>
> **Q3+Q4/Experiments and reporting.** The total number of test samples per domain is reported in the table below.
> We will add this information to the appendix for completeness.
>
> | Domain   | Elec  | Auto  | Office | Baby | Pet  | Music | Sports | Goodreads |
> |----------|-------|-------|--------|------|------|--------|--------|-----------|
> | Test Samples | 58,552 | 32,398 | 6,615 | 4,570 | 17,579 | 2,479 | 18,000 | 19,502 |
>
> For the relation‐validation prediction in Section 4.3.3, we follow a **6:2:2 split** of the above samples into train/validation/test. Each relation instance is constructed with a **1:1 positive/negative ratio**, ensuring that the classifier is trained and evaluated under balanced conditions. We will clarify these details in the revised version.We will also include the Mean Reciprocal Rank (MRR) for the relation-guided retrieval task in the revision;  the MRR table is provided in the next response.
>
> **Q5 / Clarity and terminology.** The natural-language relation descriptions are generated by clustering users based on the categories of items they purchase, and summarizing each cluster’s dominant preferences into short text. For example:
> *“Co-purchased by people who strongly prefer Tennis & Racket, while also buying Sports, Outdoor Recreation, Exercise & Fitness, and Fan Shop items…”* These summaries describe the behavioral patterns of users linking the two items and serve as the semantic context for relation conditioning. We will clarify this process more explicitly in the revised manuscript. All relation descriptions are provided in **Appendix A.2**.
>
> **Q6/Experiments and reporting.** Section 4.4 focuses on the effect of relation-conditioning strength on our relation-aware tasks. Since RCML updates the encoder toward this objective, preserving CLIP’s original zero-shot scores is not a design target and is outside the scope of our evaluation. We will clarify this point in the revision.
>
>
> **Q7/Experiments and reporting.** Yes—the CLIP baseline uses the **OpenAI ViT-B/32** weights. All baseline results are obtained using `CLIPModel` and `CLIPProcessor` from HuggingFace Transformers, which load the official OpenAI checkpoints, not OpenCLIP/LAION variants. We will add this clarification to the manuscript.

---

> ### Author Response · Authors · 2025-11-28
> **Part 2**
>
> **Different CLIP baseline.** Thank you very much for the insightful suggestion. We followed your idea and evaluated the relation-based CLIP baseline. The results below directly compare the three methods in a single table.
>
> **Comparison of all methods**
>
> | Method                       | Sports Hit@5 | Sports MRR | Office Hit@5 | Office MRR |
> |------------------------------|--------------|------------|---------------|------------|
> | Relation-based CLIP baseline | 0.2356       | 0.1718     | 0.2491        | 0.1790     |
> | Original CLIP baseline       | 0.4333       | 0.2998     | 0.3946        | 0.2608     |
> | RCML (proposed)              | 0.6449       | 0.4416     | 0.4922        | 0.3319     |
>
> As the table shows, the relation-based similarity (zA·zr)*(zB·zr) provides a meaningful relation-aware baseline, but its performance is lower than the original CLIP similarity, likely because the projection onto the relation direction removes much of the modality-specific signal. RCML substantially outperforms both baselines, confirming the benefit of explicit relation conditioning. We will include this baseline in the revised manuscript.
>
> **Table for Q3: MRR (%) for Relation-Guided Retrieval on 8 datasets using five similarity measures.
> Bold numbers indicate the best performance in each dataset.**
>
> | Similarity | Elec | Auto | Office | Baby | Pet | Music | Sports | Goodreads |
> |------------|------|------|--------|------|------|--------|---------|-----------|
> | CLIP (TT) | 25.97 | 25.95 | 26.08 | 25.14 | 26.72 | 26.26 | 29.98 | 26.75 |
> | CLIP (II) | 22.29 | 21.69| 21.86 | 21.16 | 22.56 | 22.50 | 24.83 | 22.14 |
> | CLIP (TI) | 23.22 | 22.85 | 23.01 | 21.20 | 23.62 | 23.52 | 27.17 | 22.08 |
> | CLIP (IT) | 23.52 | 23.36 | 23.50 | 22.01 | 24.09 | 23.72 | 27.40 | 23.42 |
> | CLIP (AVG) | 25.60 | 25.33 | 25.48 | 24.48 | 26.22 | 25.91 | 30.40 | 26.61 |
> | DeCLIP (TT) | 20.40 | 20.16 | 22.22 | 20.78 | 21.11 | 21.11 | 21.42 | 20.14 |
> | DeCLIP (II) | 17.85 | 17.99 | 18.84 | 18.63 | 18.11 | 19.85 | 18.48 | 18.39 |
> | DeCLIP (TI) | 17.27 | 16.94 | 17.40 | 17.81 | 17.02 | 17.81 | 17.59 | 17.32 |
> | DeCLIP (IT) | 17.84 | 16.81 | 17.31 | 17.19 | 16.99 | 18.35 | 17.52 | 17.55 |
> | DeCLIP (AVG) | 19.95 | 19.98 | 21.65 | 21.42 | 20.85 | 21.50 | 21.38 | 21.04 |
> | UniCL (TT) | 18.24 | 18.99 | 18.96 | 18.33 | 18.53 | 19.74 | 18.67 | 18.43 |
> | UniCL (II) | 21.97| 20.88 | 25.19 | 21.47 | 24.40 | 24.66 | 24.87 | 24.83 |
> | UniCL (TI) | 17.91 | 18.06 | 17.20 | 17.10 | 17.06 | 17.68 | 17.33 | 17.38 |
> | UniCL (IT) | 17.13 | 17.34 | 17.10 | 16.54 | 17.27 | 17.40 | 17.61 | 17.36 |
> | UniCL (AVG) | 21.30 | 21.02 | 24.43 | 21.29 | 23.82 | 24.63 | 23.87 | 21.44 |
> | SigLIP (TT) | 27.22 | 27.22 | 29.41 | 27.35 | 26.72 | 27.54| 32.99 | 27.32 |
> | SigLIP (II) | 25.44 | 24.61 | 26.78 | 23.82 | 22.56 | 25.69 | 29.08 | 26.55 |
> | SigLIP (TI) | 25.43 | 25.18 | 27.41 | 24.36 | 23.62 | 25.77 | 31.31 | 26.93 |
> | SigLIP (IT) | 25.31 | 25.03 | 27.92 | 24.56 | 24.09 | 25.61 | 32.03 | 26.46 |
> | SigLIP (AVG) | 27.40 | 27.02 | 29.45 | 26.49 | 26.22 | 27.73 |33.42 | 27.59 |
> | ImageBind (TT) | 27.78 | 27.69 | 27.80 | 27.75 | 28.46| 28.11 | 29.17 | 27.69 |
> | ImageBind (II) | 26.63 |  **25.62** | 25.77 | **25.18** | 26.40 | 26.87 | 26.87 | 26.04|
> | ImageBind (TI) | 26.16 | **26.10** | 26.23 | 25.27 | 26.82 | 26.51 | 27.54 | 25.94 |
> | ImageBind (IT) | 26.00 | 25.77 | 25.88 | 24.47 | 26.49 | 26.30 | 27.23 | 26.75 |
> | ImageBind (AVG) | 28.37 | 27.78 | 27.89 | 27.31 | 28.60 | 28.69 | 29.29 | 27.04 |
> | RCML (TT) | **32.95** | **30.34** | **33.19** | **29,40** | **35.96** | **34.94** | **44.16** | **32.14** |
> | RCML (II) | **28.23** | 25.16 |**28.65** | 24.15 | **28.92** | **28.01** | **31.06** | **29.11** |
> | RCML (TI) | **27.53** | 25.32 | **30.00** | **26.08** | **30.66** | **30.80** | **34.85** | **29.41** |
> | RCML (IT)| **32.28**| **27.64** | **30.60** | **26.44**| **34.02** |**28.97**| **42.56**| **30.13**|
> | RCML (AVG) | **32.25**| **29.38** | **33.38** |**28.93** | **35.69** | **35.14**| **42.93** | **32.09**|

---

> > ### Comment · Area_Chair_NaC7 · 2025-11-28
> >
> > Dear Reviewer,
> >
> > Please make sure you read the authors' response and engage with them in the discussion before the end of the discussion period on **Dec 03 '25 09:00 PM UTC**. This is a hard deadline.
> >
> > Thank you for supporting quality peer review at ICLR.
> >
> > AC

---

### Meta-Review · Area_Chair_CC98 · 2026-01-06

**Summary:**

This paper received consistent negative review ratings. Although the reviewers consider the idea of introducing relations as a condition to modulate CLIP embeddings makes sense, the overall contribution appears incremental. The paper is not well prepared, e.g., missing the reference indexes in quite a few places. The comparisons with baselines is somewhat unfair, since the proposed method is trained with relation data, while the baselines were not. R3’s request on fine-tuning CLIP with relation data could be meaningful via some simple architectures, such as MLP or cross-attention. Unfortunately, the authors deem it is unnecessary in the rebuttal. Thus, this AC can’t recommend acceptance and hopes the detailed review comments will help strengthen the work.

**Reviewer Concerns:**

See above.

**Reviewer Scores:**

One reviewer had indicated that they would not change. The first one might upgrade the rating since most of his detailed questions were answered in the rebuttal. However, others are unlikely to change.

---

### Decision · Program_Chairs · 2026-01-26

Reject